# Microbicidal Activity of Extract *Larrea tridentata* (Sessé and Moc. ex DC.) Coville on *Pseudomonas syringae* Van Hall and *Botrytis cinerea* Pers

**DOI:** 10.3390/microorganisms13051055

**Published:** 2025-04-30

**Authors:** Diego Rivera-Escareño, Jorge Cadena-Iñiguez, Dalia Abigail García-Flores, Gerardo Loera-Alvarado, Lizeth Aguilar-Galaviz, María Azucena Ortega-Amaro

**Affiliations:** 1Colegio de Postgraduados, Innovación en Manejo de Recursos Naturales, Campus San Luis Potosí, Iturbide 73, Salinas de Hidalgo C.P. 78600, SLP, Mexico; escareno.diego@colpos.mx (D.R.-E.); garcia.dalia@colpos.mx (D.A.G.-F.); gerardo.loera@colpos.mx (G.L.-A.); aguilar.lizeth@colpos.mx (L.A.-G.); 2Coordinación Académica Región Altiplano Oeste, Universidad Autónoma de San Luis Potosí, Carretera Salinas-Santo Domingo 200, Salinas de Hidalgo C.P. 78600, SLP, Mexico; azucena.ortega@uaslp.mx

**Keywords:** phytochemicals, crop diseases, botanical extracts, sustainable control

## Abstract

Due to their secondary metabolite content, plant extracts are an alternative method for controlling pathogenic organisms in agriculture and post-harvest operations. *Botrytis cinerea* and *Pseudomonas syringae* are among the causative agents of diseases and losses in agricultural production. The species *Larrea tridentata* is abundant in the arid and semi-arid zones of Mexico and has no defined use; however, it contains secondary metabolites with microbicidal potential that could aid in biological control and enhance its harvest status. Growth inhibition (halo) of *B. cinerea* and *P. syringae* was evaluated by applying alcoholic extract of *L. tridentata* leaves at doses of 50, 100, 250, 500, 750, 1000, and 2000 µg mL^−1^ in vitro, using poisoned medium and potato dextrose agar for the fungus and the agar well method for the bacteria, in a completely randomized design with five replicates. The flavonoids quercetin, apigenin, narigenin, kaempferol, and galangin were identified as possible agents of microbicidal activity. The extract inhibited the growth of *B. cinerea* from 100 µg mL^−1^ and completely inhibited it with 1000 and 2000 µg mL^−1^. For *P. syringae*, inhibition was observed from 250 µg mL^−1^, demonstrating that the higher the concentration, the greater the growth inhibitory effect. The secondary metabolite content of the *L. tridentata* extract is sufficient to have an impact on microorganisms with economic impact in agriculture.

## 1. Introduction

Crop diseases are a recurring problem for producers because they affect their yield [1]. The use of synthetic agrochemicals is common for disease control in agriculture, but it has a negative impact on the environment [2]. In addition to increasing production costs, they can affect the health of producers and populations near the areas where they are applied [3,4]. Among the pathogens that cause diseases in a wide variety of hosts, the *Pseudomonas syringae* and *Botrytis cinerea* stand out, causing considerable economic losses [5]. Both microorganisms attack crops and agricultural products due to their easy dispersal through air and rain. Traditional treatments induce resistance to the active ingredients, increasing the problem in the medium term [6,7]. Therefore, botanical extracts are a viable option for crop disease control thanks to their microbicidal action and are often used in sustainable agricultural practices [8,9,10]. They generally cause less environmental damage due to their rapid degradation, which reduces the risk of products containing residues harmful to consumers [11]. Therefore, the use of endemic plants as microbiocidal agents for crop disease control should be given greater importance. Such is the case of *Larrea tridentata*, which contains secondary metabolites (Sm) that can control pathogens due to their biological activity as antimicrobials, antiparasitics, antifungals, and repellents [12,13,14,15].

## 2. Materials and Methods

### 2.1. Collection of Plant Material

The collection was carried out in an area of microphyllous desert vegetation in Zacatecas, Mexico (22°37’49″ N 101°52′58.2″ W), during the dry season. Random sampling was carried out with random points using the QGIS program Q3.16.15 (Hannover, Germany). The upper third of the branches of *Larrea tridentata* was collected, so as not to affect the integrity of the plant [16].

### 2.2. Extraction

The biomass was purified by separating the leaves from the stems. It was dried at room temperature under shade for seven days, then the particle size was homogenized by pulverizing the plant material with an HC-700Y mill (Grinder, Zhengzhou, China) and sieving it with a #20 sieve. To obtain the extract, the plant was macerated in a 500 mL flask with 50 g of plant material and 250 mL of 80% ethanol for seven days at room temperature, protected from light, and covered with aluminum foil. Plant residues were subsequently removed by vacuum filtration. The extract was concentrated by vacuum distillation using a rotary evaporator (RE100-PRO, DLAB, Los Angeles, CA, USA) at 40 °C and 120 rpm, until it reached a semi-liquid state. It was then placed in a drying oven (ECOSHEL, Pharr, TX, USA) at 40 °C for seven days, and a resinous extract was obtained.

### 2.3. Dilution Preparation

For the stock, 0.2 g of the extract was weighed into a 50 mL falcon tube, 0.5 mL of ethanol (1%) was added, and the mixture was allowed to stand for 10 min to facilitate extract dilution. An amount of 49.5 mL of distilled water was subsequently added. The mixture was then placed in an ultrasonic bath for 5 min and stirred in a Vortex-Genie™ 2 (Scientific Industries, 80 Orville Drive, Suite 102, Bohemia, NY, USA). This dilution gave a concentration of 4000 µg mL^−1^, from which concentrations of 50, 100, 250, 500, 750, 1000, and 2000 µg mL^−1^ were obtained. This process was performed for each repetition of the tests on the microbicidal effect.

### 2.4. Inoculum Preparation

The inocula were provided by the laboratories of the National Institute of Forestry, Agriculture and Livestock Research, Mexico city, Mexico, and the Potosino Institute of Scientific and Technological Research (IPICyT), San Luis Potosí, Mexico. The *B. cinerea* strain B05.10 was cultured on potato dextrose agar (PDA) medium and incubated at 25 °C for four days before use. The *Pseudomonas syringae* DC3000 strain was replated onto Luria-Bertani (LB) agar and incubated at 37 °C for 48 h before use.

### 2.5. Microbicidal Effect on B. cinerea

The poisoned medium method [17] was used to evaluate the microbicidal effect, with five replicates performed for each concentration. For each replicate, 20 mL of poisoned medium was used. Therefore, 100 mL of this medium was prepared per concentration by adding 50 mL of distilled water and 3.9 g of potato dextrose medium (PDA, DIBICO^®^, San Nicolás de los Garza, Mexico) to a flask for 100 mL. The mixture was then diluted on a hot plate at 120 °C and autoclaved at 120 °C for 15 min (Lab Companion, model ST-105 GP, Daejeon, Republic of Korea). Another 50 mL of sterilized water was used to dilute the extract.

The medium and the dilution were mixed vigorously to homogenize the medium, poured into 90 × 10 plastic Petri dishes, and left to stand for 4 h to solidify properly. This process was performed for concentrations of 50, 100, 250, 500, 750, 1000 and 2000 µg mL^−1^, in addition to including a negative control (c−) of PDA agar without extract concentrations and a positive control (c+) with the commercial fungicide Ridomil Gold Bravo (metalaxyl-M: methyl N-(2,6-dimethyl phenyl)-N-(2-methoxy-acetyl)-D-alaninate at 4%, and 64% mancozeb: ethylene bis-dithiocarbamate of manganese and zinc). PDA medium discs were inoculated with the fungus *Botrytis cinerea* after four days of growth and incubated at 25 °C (ECOSHEL, Pharr, TX, USA). Micellar growth was measured in two cross-sectional areas until the control covered the entire Petri dish.

### 2.6. Microbicidal Effect on P. syringae

The agar well method [18] was used. The *P. syringae* inoculum was collected with a bacteriological loop and suspended in a test tube with 5 mL of 0.85% saline solution. Its absorbance was measured in a spectrophotometer (Thermo SCIENTIFIC, model GENESYS 10s Vis, Labconco Corporation, 8811 Prospect Ave., Kansas City, MO, USA) at 625 nm. It was adjusted to 0.5 on the McFarland scale, giving an absorbance range of 0.08 to 0.13 [19]. LB culture medium was prepared using 10 g of casein peptone (DIBICO, San Nicolás de los Garza, México), 10 g of NaCl, and 15 g of bacteriological agar (DIBICO) in proportion to 1.0 L of culture medium. With 10 µL of this strain, Petri dishes containing LB culture medium were inoculated using a Drigalsky loop (Sociedad Importadora Liplata, El Juncal 110, 8720025 Quilicura, Región Metropolitana, Chile), ensuring an even application over the entire Petri dish. Subsequently, four perforations were made on the agar with a 7 mm hole punch. Two drops of agar were placed in the perforations, and 100 µL of the extract dilutions of 100, 250, 500, 750, 1000 and 2000 µg mL−1 were added in triplicate for each dose. Sterile distilled water was used as a negative control. The mixture was then added to 37 °C for 18 h to measure two radii of the inhibition zone analyzed with a completely randomized design and analysis of variance (ANOVA) (*p* < 0.05).

### 2.7. Total Phenol Content and Flavonoid Content

To evaluate the total phenol content of *L. tridentata* extract, the Folin–Ciocalteu technique [18] was used. A concentration of 10% Folin’s reagent and 7.5% NaCO were used. Concentrations of 100, 200, 300, and 500 µg mL^−1^ were evaluated. A total of 300 µL of the extract concentration to be evaluated was added to test tubes, followed by 1.5 mL of Folin’s reagent and 1.2 mL of CaCO_3_. The sample was incubated in a thermobath (Riossa, model B-80, Equipos de medición industrial, Calle de hule 12 a Col. Cd cuauhtemoc seccion llanetes, 55067 Ecatepec Estado de Mexico, Mexico) at 50 °C for 40 min. Absorbance was then measured in a spectrophotometer with absorbance at 760 nm. The same proportions of Folin’s reagent and sodium carbonate were used as blanks. A calibration curve was created with gallic acid (0.0093 to 0.15) to determine the milligrams of phenols equivalent to gallic acid [20].

The flavonoid content was determined using the aluminum chloride method. An amount of 2 mL of distilled water was poured into a test tube, followed by 500 µL of the extract dose to be evaluated. An amount of 150 µL of 10% AlCl_3_ was added. The tube was allowed to stand for 6 min, and then 2 mL of 1 M NaOH was added. The volume was made up to 5 mL with distilled water, and the absorbance was measured in a spectrophotometer at 510 nm. The experiment was performed in triplicate. To estimate the amount of mg of flavonoids equivalent to quercetin, a calibration curve was created, with concentrations ranging from 0.038 to 5 mg mL^−1^ [21].

### 2.8. Flavonoid Identification (HPLC)

Compound identification was performed using high-performance liquid chromatography using Agligent equipment. The equipment consisted of a manual injector, a degasser, a quaternary pump (Agligent 1200 series, 5301 Stevens Creek Blvd, Santa Clara, CA, USA) with a Zorbax Eclipse C18 column at room temperature, and a variable wavelength detector [Agligent 1200 series G1314B/G1314C (SL)] using the method described in [22]. The UV detector was set to a wavelength of 280 nm, and the extract and standards (apigenin, galagenin, kaempferol, quercetin, and apigenin) were filtered through a 0.2 µm membrane filter. The mobile phase consisted of acetonitrile and 0.3% acetic acid in water, with gradients of 30% and 70% from 0 to 2 min, 50% and 50% from 2 to 11 min, 70% and 30% from 11 to 17 min, up to 100% and 0% acetonitrile and 0.3% acetic acid, respectively, with a flow rate of 1 mL min^−1^. The data were analyzed using Statistica 7 software with a repeated measures ANOVA over time, followed by a Tukey test (*p* < 0.05).

## 3. Results

### 3.1. Extract Obtained

The yield achieved in this study was 23.4% based on the dry weight of the plant (Table 1).

### 3.2. Fungicidal Effect on B. cinerea

The negative control with the fungus *B. cinerea* covered the entire plate in five days, which is why these were the measurements taken for this study. Concentrations of c−, c+, 50, 100, 250, 500, 750, 1000, and 2000 µg mL^−1^ were found to have significant differences at all-time points. After analyzing the data using Statistica 7 software with a repeated measures ANOVA, followed by a Tukey post hoc test (*p* < 0.05) for each time point, it was determined that at 24 h, all treatments showed similar growth, except for concentrations of 1000 and 2000 µg mL^−1^ (Figure 1a). At 48 h, the differences between concentrations were notable. The negative control was similar to the lowest concentration (50 µg mL^−1^), while the positive control was similar to the concentration of 100 µg mL^−1^, while concentrations of 1000 and 2000 µg mL^−1^ still showed no growth (Figure 1b).

During the 72 h time point, the negative control remained the same at the concentration of 50 µg mL^−1^, while the positive control was different from the negative, but similar to the concentrations of 50 and 100 µg mL^−1^. The concentration of 1000 µg mL^−1^ turned out to be equal to the concentrations of 750 and 2000 µg mL^−1^ (Figure 1c). For the 96 h time point, the statistical differences and equalities were identical to those of the 72 h time point, with the exception that the equality between 750 and 1000 µg mL^−1^ did not occur at this time point (Figure 1d) but was recorded again at the 120 h time point (Figure 1e), where the differences and similarities were the same as at the 72 h time point. For the last time point (120 h), the negative control turned out to be equal to the dose of 50 µg mL^−1^ and different from the rest, the positive control was equal with the doses of 50 and 100 µg mL^−1^ and different from the negative control and the other doses.

The 50 µg mL^−1^ dose was statistically equal to the controls and the 100 µg mL^−1^ dose. The 500 µg mL^−1^ dose was statistically equal to the 250 and 750 µg mL^−1^ doses, while the 750 concentration was statistically equal to the 250, 500, and 1000 µg mL^−1^ doses but different from the 2000 µg mL^−1^ dose. The 1000 µg mL^−1^ concentration showed equal inhibition to the 750 and 2000 µg mL^−1^ doses (Figure 1e). The 1000 and 2000 µg mL^−1^ concentrations had a fungicidal effect on two of five plates and three of five, respectively. The above suggests a fungistatic effect proportional to the increase in concentrations, such that, starting in the first 24 h, *B. cinerea* growth was inhibited at low concentrations. It is noteworthy that at 120 h, *B. cinerea* inhibition registered 96% inhibition of mycelial growth, which was superior to the positive control (Table 2).

### 3.3. Bactericidal Effect on P. syringae

For the analysis of the bactericidal effect on *P. syringae*, a one way ANOVA (*p* value 0.05) was performed, showing that 50 and 100 µg mL^−1^ were statistically equal to the negative control. However, the bactericidal effect of the extract was recorded from the dose of 250 µg mL^−1^, inhibiting growth (halo) as the concentration increased, without registering statistical differences (Figure 2).

### 3.4. Total Phenol and Flavonoid Content

The total phenol content is expressed in two forms: milligrams of total phenols equivalent to gallic acid per gram of extract (FTEAG mg gE^−1^) and milligrams of total phenols equivalent to gallic acid per gram of dry matter (FTEAG mg gD^−1^) (Table 3).

### 3.5. Compound Identification and Quantification (HPLC)

Using high-performance liquid chromatography, some of the common flavonoids contained in the *L. tridentata* extract were identified, highlighting quercetin (Q) with a retention time of 3500, apigenin (A) with a retention time of 4100, naringenin (N) with a retention time of 4200, kaempferol (K) with a retention time of 4400, and galagenin (G) with a retention time of 6700 (Figure 3)

## 4. Discussion

Species of the *Larrea* genus, such as *L. divaricata* and *L. nitida*, have been shown to have significant bactericidal potential [22,23]. Other studies have tested the efficacy of *L. tridentata* extracts against bacteria of human medical importance [13], attributing the biological activity to the β-lactam content, an active molecule in common antibiotics, such as penicillin. Ref. [22] showed that *L. tridentata* extracts are comparable with commercial antibiotics for the control of bacteria of clinical interest. Other research has shown that some bacteria that affect animal health [23] are controlled by the presence of nor 3′-demethoxyisoguaiacin in *L. tridentata* extracts, and it is also effective against bacteria resistant to common antibiotics associated with bovine mastitis.

The efficacy of *L. tridentata* extracts against other antibiotic-resistant bacteria of agricultural importance has been demonstrated [14]. Similarly, flavonoids present in *L. tridentata* extracts presented a relevant bactericidal effect [24,25,26].

Among the flavonoids, apigenin, kaempferol, naringenin, galangin, and quercetin have been identified, all of which possess bactericidal properties on a wide range of species, such as *Escherichia coli* and *Staphylococcus aureus* [27,28,29,30,31]. They have also recorded relevant activity against fungi such as *Fusarium solani*, *Fusarium oxysporum*, *Trichoderma* spp., *Penicillium notatum*, and *Aspergillus niger*, among others [32,33,34].

This finding is very important, considering the ban on antibiotics in agriculture and the treatment of farm animals, as their use is directly associated with the transfer of resistant bacteria to the animals themselves and to humans. The Food and Agriculture Organization of the United Nations [35] called for urgent measures to address the health crisis represented by antimicrobial resistance (AMR), including antibiotics, antivirals, antifungals, and antiparasitics. According to a study published in *The Lancet*, it is estimated that in 2019 [36], around five million deaths related to antimicrobial resistance occurred.

Therefore, the use of botanical extracts for disease control is gaining importance, as they reduce the possibility of microorganisms developing resistance, due to their content of different DMs as active ingredients [37]. A determination of the phenol content in a methanolic extract of *L. tridentata* obtained a value of 211.18 mg gE^−1^ [38]. Ref. [39] mention that the content of compounds extracted from a plant is greatly influenced by the extraction solvents and their concentration. The content of secondary metabolites in *L. tridentata* varies depending on the organ from which the extraction is made [40].

Phenols are compounds with inherent antioxidant activity [41]. Flavonoids are also linked to antioxidant capacity [42], so plants with high phenolic content have potential for drug development, as this is related to their antioxidant capacity [43]. In this regard, plants grown in arid areas contain Sm content that can aid in the control of phytopathogens in crops and postharvest products [44]. Additional metabolite content has been reported in *L. tridentata* extracts, with nordihydroguaiaretic acid (NDGA) being particularly notable, reported as one of the most potent antioxidants, with significant bioactive capacities [45,46].

The different retention times for quercetin and kaempferol demonstrate that the extraction method affects retention times [22]. Retention times can vary depending on measurement conditions and equipment. Another study shows that the use of plant extracts such as essential oils can help reduce the use of agrochemicals for disease control by mixing these compounds with the active ingredients of commercial fungicides [47].

The treatment of plant pathogenic bacteria with conventional agrochemicals often attracts problems such as resistance and significant economic costs [6]. The use of plant extracts for the control of pseudomonas has been evaluated using methanolic extracts of *Zingiber officinale*, with a 14 mm reduction; *Opuntia ficus-indica*, with a 13 mm reduction; *Bryophyllum pinnatum*, with a 12 mm reduction; *Syzygium romaticum*, with a 16 mm reduction; *Gingiber officinale*, with a 10 mm reduction; *Syzygium aromaticum*, with a 14 mm reduction; and *Curcuma longa*, with a 0 mm reduction.

*L. tridentata* extracts at doses of 1000 and 2000 µg mL^−1^ showed inhibition of 19.27 mm and 20.15 mm, respectively, using organic solvents [48]. Therefore, plant extracts prove to be a viable option for controlling the excessive resistance to antibiotics presented by phytopathogenic bacteria [49].

The inhibition of *Pseudomonas syringae* pv. *tobacco* and *Pseudomonas syringae* pv *tomato* by industrial hemp extracts was evaluated, where, at a concentration of 12.5 mg mL^−1^, an inhibition halo of 33 mm was obtained, and when using 3.13 mg mL^−1^, an inhibition halo of 28.7 mm was recorded. The results obtained in the present study indicate that the inhibition achieved by the ethanolic extracts of *L. tridentata* are sufficient for the control of *P. syringae* [50].

The phytopathogenic fungus *B. cinerea* has developed resistance to different synthetic fungicides that previously represented a solution to the disease in crops [51,52]. Different biological control alternatives for the control of *B. cinerea* have resulted in solutions to the problem of resistance to synthetic fungicides, among which is the use of botanical extracts [53]. Plant extracts are a viable option for increasing shelf life, since they protect fruits and vegetables from postharvest diseases [54,55,56]. Extracts of *L. tridentata* have inhibited phytopathogenic fungi such as *Fusarium oxysporum*, *Fusarium solani*, and *Rhizoctonia solani* [45]. Other species of the genus *Larrea*, such as *L. cuneifolia*, showed their efficiency in the control of postharvest diseases [57]. Therefore, the properties of *L. tridentata* extracts can be used to increase the shelf life of fruits and vegetables by controlling phytopathogens such as *B. cinerea*.

The fungicidal effect of blueberry branches and leaves (mixture) and avocado seeds at 5, 25, and 50 mg mL^−1^ was evaluated on the inhibition of spore germination and mycelial growth of *Botrytis* sp., recording 55% spore inhibition at 24 h of incubation and 19% inhibition of mycelial growth at seven days of incubation. The effect was attributed to polyphenols, as determined by the Folin–Ciocalteu method [58]. The chemical characterization of resins, global extracts, essential oils, and decoctions, as well as the biological activity of the *Larrea* genus, have been extensively studied and reported. The total phenolic content (239.5 mg EAG/g of SAO extract and 215.6 mg EAG/g of MAQ extract) and significant differences in the content of flavonoid compounds (28.44 and 15.28 mg EQ/g in EMLa SAO and in EMLa MAQ, respectively). Medium-polarity flavonoids and polar phenolic compounds have been identified, highlighting the presence of nordihydroguaiaretic acid, a biomarker of the *Larrea* genus with fungicidal and bactericidal activity [59]. Other studies report that *Lippia origanoides* essential oil inhibited 92.3% to 94.3% of the disease, while *Thymus vulgaris* recorded inhibition rates from 92.2% to 93.6%. The utilization of essential oils is better, but still lagging behind *Larrea tridentata* extracts at doses of 1000 and 2000 µg mL^−1^. The inhibition capacity of extracts of cinnamon (*Cinnamomum verum*), pepper (*Capsicum annuum*), and bay leaf (*Laurus nobilis*) against *B. cinerea* isolated from strawberry was evaluated. Complete inhibition of micellar growth was observed at 2200, 2400, and 2600 µL L^−1^ for bay leaf and pepper extracts, followed by cinnamon, which completely inhibited it at concentrations of 600 and 800 [60].

Mexico has a plant diversity of more than 25,000 species, of which around 100 have economic value [61]. Medicinal plants such as *L. tridentata* can be an option for economic development in rural areas [62], since its biological activity against pathogens that cause diseases in crops allows the design of formulations and compositions (blends) with the extracts. This can support the development of new value networks in rural communities in the semidesert that face agroclimatic limitations for conventional agriculture.

## 5. Conclusions

*L. tridentata* extract exhibits biological activity that significantly inhibits the growth of *B. cinerea* and *P. syringae,* making it a viable option for controlling plant pathogens. The utilization and revaluation of this natural resource can contribute to its conservation and may reduce the use of agrochemicals. The secondary metabolite content of *L. tridentata* suggests that future phases should include in vivo evaluations with formulations and compositions for mass use.

## Figures and Tables

**Figure 1 microorganisms-13-01055-f001:**
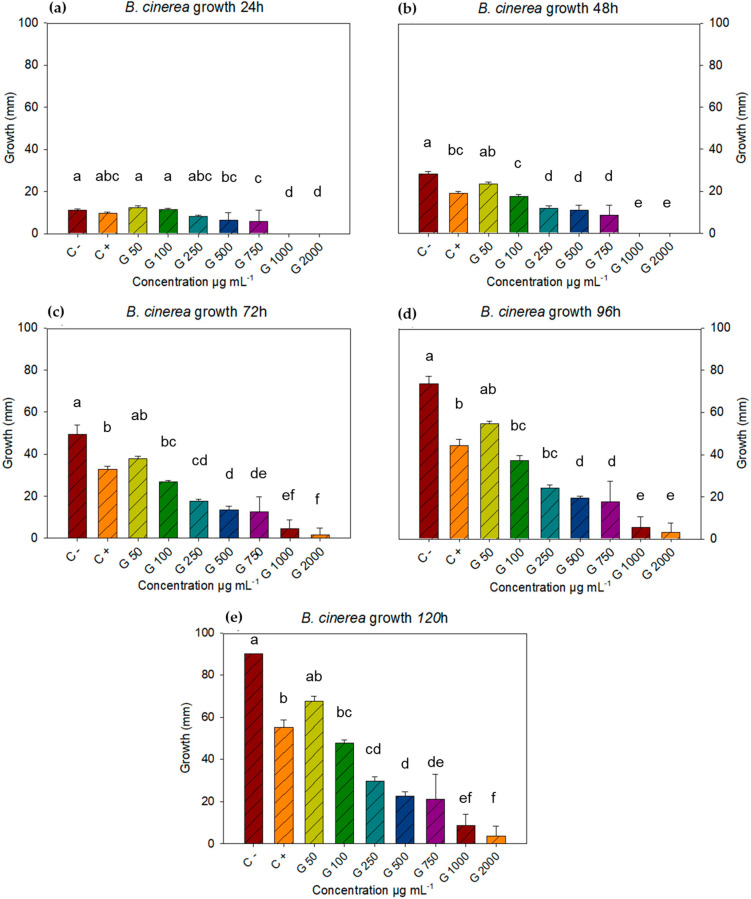
Growth of *B. cinerea* with doses of *L. tridentata* extract at 24, 48, 72, 96, and 120 h, and concentrations of 50, 100, 250, 500, 750, 1000, and 2000 µg mL^−1^, using Tukey’s test (*p* < 0.05). Average values ± standard error. ANOVA, Tukey’s test (*p* ≤ 0.05). Equal letters for each factor are significantly different.

**Figure 2 microorganisms-13-01055-f002:**
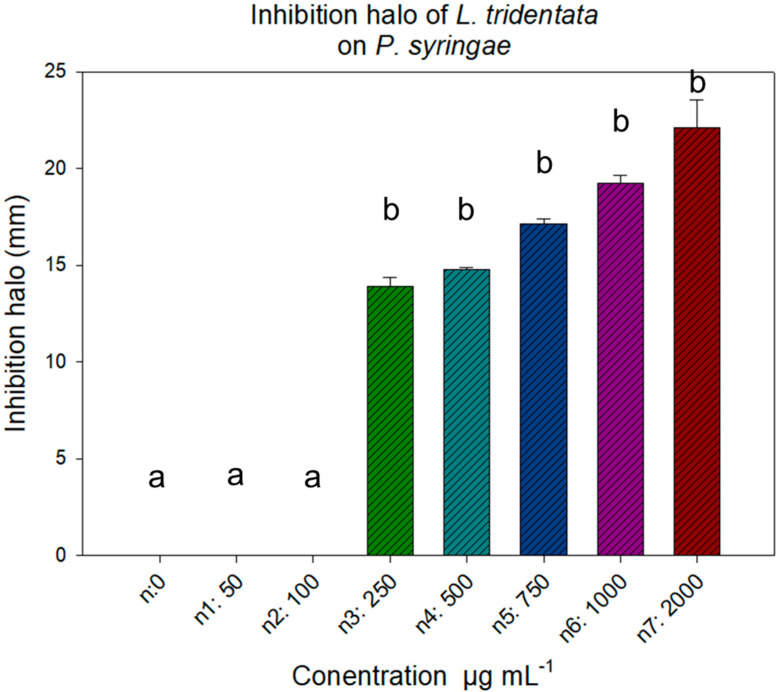
Growth inhibition of *Pseudomona syringae* with *L. tridentata* extract at 18 h of incubation. Average values ± standard error. ANOVA, with Tukey’s test (*p* ≤ 0.05). Equal letters for each factor are significantly different.

**Figure 3 microorganisms-13-01055-f003:**
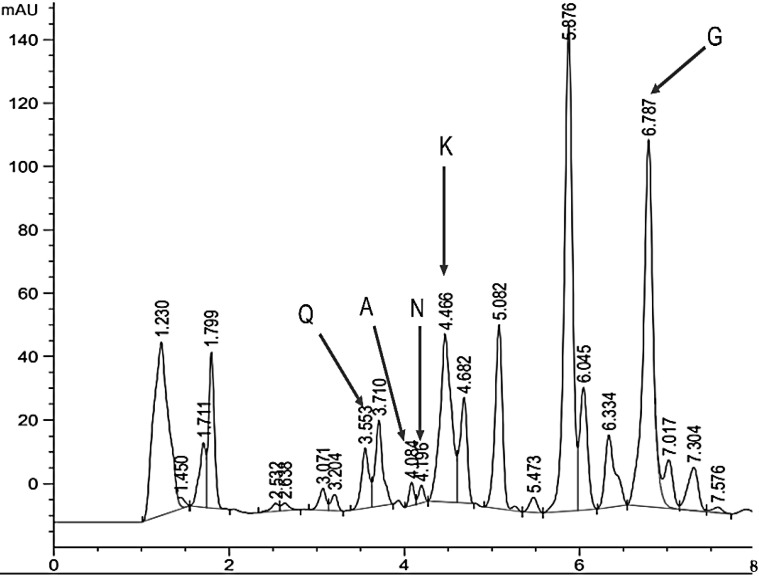
Chromatogram (HPLC) of a sample of 900 µg mL^−1^ of ethanolic extract of *L. tridentata*. Q: quercetin; A: apigenin; N: narigenin; K: kaempferol; G: galangin.

**Table 1 microorganisms-13-01055-t001:** Yield obtained from ethanolic extract of *L. tridentata*.

Extract Yield	
Dry matter weight	50.0 g
Extract obtained	11.7 g
Dry matter yield	23.4%

**Table 2 microorganisms-13-01055-t002:** Inhibition of *L. tridentata* extract on *B. cinerea* at 120 h. Average values of three replicates ± standard error.

Concentration (µg mL^−1^)	Inhibition (%)
C-	0 ± 0
C+	39.62 ± 3.81
50	24.91 ± 19.86
100	46.97 ± 1.59
250	67.1 ± 2.24
500	74.93 ± 2.56
750	76.7 ± 7.98
1000	90.42 ± 5.92
2000	96.1 ± 5.41

**Table 3 microorganisms-13-01055-t003:** Phenol and flavonoid content in *L. tridentata* extracts. Milligrams of total phenols equivalent to gallic acid per gram of extract (FTEAG mg gE^−1^).

Phenol Content EAG	
FTEAG mg gE^−1^	291.02
FTEAG mg gMS^−1^	73.92
FlEQ mg gE^−1^	598.27
FlEQ mg gMS^−1^	153.40

FTEAG: total phenols equivalent to gallic acid. FlEQ: flavonoids equivalent to quercetin. gE: grams of extract. gMS: grams of dry matter.

## Data Availability

The original contributions presented in this study are included in the article. Further inquiries can be directed to the corresponding author.

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
