# Peer review of "Microbicidal Activity of Extract Larrea tridentata (Sessé and Moc. ex DC.) Coville on Pseudomonas syringae Van Hall and Botrytis cinerea Pers"

_microorganisms, 2025, doi:10.3390/microorganisms13051055_

Round 1

Reviewer 1 Report

Comments and Suggestions for Authors

The topic of the work is current and interesting, however, the authors should carry out, in addition to in vitro tests, biological tests, including preventive and curative control of plant diseases caused by pathogens. It would also be necessary to verify any phytotoxic effects due to the concentrations of the plant extracts.

Comments on the Quality of English Language

Can be improved.

Author Response

Dear Reviewer,

Please refer to the file attached.

Reviewer 2 Report

Comments and Suggestions for Authors

The manuscript "Microbicidal activity of Larrea tridentata extract (Sessé & Moc. ex DC.) Coville on Pseudomonas syringae and Botrytis cinera" by Diego Rivera-Escareño et al. presents a report on the use of Larrea tridentata as an antimicrobial agent. The article as a whole is systematized and valuable, but there are a number of comments on it.

  1. The "agar well" method is described superficially: the number of repetitions is not specified, control data (positive/negative) is not provided, there is no mention of statistical tests for these data (although they are later mentioned in pp. 179-184). This reduces the reproducibility of the experiment.
    2. The concentration of 500 micrograms/ml is indicated twice in the list of doses applied to B. cinerea: “...concentrations 50, 100, 250, 500, 500, 750, 1000 and 2000 µg mL⁻1...”. This is an editorial error that needs to be corrected, as it may cause confusion when interpreting the results.
    3. The authors claim "statistical differences" between different concentrations, but they do not accompany this with p-value data, F-statistics, or ANOVA details. Also, descriptions like “statistically equal" between groups without clear criteria create great confusion in interpreting the results. It is a great request to carry out statistical processing and analyze all the data with this in mind.
    4. Although the content of flavonoids is indicated and their types are listed (quercetin, apigenin, etc.), the authors do not associate these data with activity against specific pathogens. There is no attempt to establish a possible causal relationship between phytanalysis and biotests.
    5. The conclusions state that the extract may be a "viable option for controlling plant pathogens", but it does not specify which next step: field use, drug development, or toxicological testing. Conclusions and discussions need to be specified and expanded.
    6. Although the extract preparation method is described in detail, there is no data on the quality control of the extract between batches, there is no information on the validation of the method (for example, repeated testing on different cultures or batches of extract), which undermines confidence in the applicability of the results obtained.
    7. There are editorial errors, for example, in line 252 "The evaluation of the inhibition of P. syringae pv. Tobacco and P. syringae pv Tomato." The species names should be given in accordance with the requirements of the journal.
    8. It is unclear what the specific novelty of the study is and how the authors intend to use this data. The fact that plant extracts have an antibacterial effect has been known for a long time, so you should specify how this study differs from others in more detail.   9.There is also a question about the title of the work. Shouldn't the initials of the plant discoverers be placed after its mention, rather than after the word "extract"

Correcting these inaccuracies would improve the quality of the article.

Author Response

(The authors gave the same response as above.)

Round 2

Reviewer 1 Report

Comments and Suggestions for Authors

We have to consider that in vivo results may be different from those obtained under controlled conditions.

Author Response

Comments: We have to consider that in vivo results may be different from those obtained under controlled conditions.

Reply: The authors agree with the reviewer and must keep in mind that in vivo results may differ from those obtained under controlled conditions. However, in vitro evaluations and subsequent treatment with Koch's postulates can help us with field evaluations.

Reviewer 2 Report

Comments and Suggestions for Authors

First of all, I don't see any answers from the authors to the questions, except for the corrected file. Secondly, the authors have not yet carried out statistical processing of the results.

Author Response

Comments: First of all, I don't see any answers from the authors to the questions, except for the corrected file. Secondly, the authors have not yet carried out statistical processing of the results.

Reply: The Figures have been corrected, highlighting the difference from the mean with lowercase letters for each bar, based on Tukey's test and p < 0.05.
